# Mamba State-Space Models Can Be Strong Downstream Learners

## Abstract

Mamba [22] state-space models (SSMs) have recently outperformed state-of-the-art (SOTA) Transformer large language models (LLMs) in various tasks and been widely adapted. However, Mamba's downstream learning capabilities remain either unexplored–e.g., mixed-precision (MPFT) and parameter-efficient fine-tuning (PEFT)–or under-evaluated–e.g., in-context learning (ICL). For the latter, recent works [45, 19] reported Mamba's ICL rivals SOTA Transformer LLMs using non-standard benchmarks. In contrast, we show that on standard benchmarks, pretrained Mamba models achieve only 38% of the ICL performance improvements (over zero-shot) of comparable Transformers.

Enabling MPFT and PEFT in Mamba architectures is challenging due to recurrent dynamics and highly customized CUDA kernels, respectively. However, we prove that Mamba's recurrent dynamics are robust to small input changes using dynamical systems theory. Empirically, we show that performance changes in Mamba's inference and fine-tuning due to mixed-precision align with Transformer LLMs. Furthermore, we show that targeting key memory buffers in Mamba's customized CUDA kernels for low-rank adaptation regularizes SSM parameters, thus achieving parameter efficiency while retaining speedups. We show that combining MPFT and PEFT enables up to 2.15 times more tokens-per-second and 65.5% reduced per-token-memory compared to full Mamba fine-tuning, while achieving up to 81.5% of the ICL performance improvements (over zero-shot) of comparably fine-tuned Transformers.

## 1 Introduction

Innovating on previous state-space models (SSMs) [23, 11], Mamba [22] has been recently proposed as an accurate, sub-quadratic alternative to Transformer large language models (LLMs). Mamba was initially shown to greatly outperform comparable Transformer LLMs [5] across a large number of standard natural language benchmarks. Subsequently, pretrained Mamba models have been widely adapted across different data modalities [42, 65, 36, 46, 37], tasks [60, 62, 48, 63, 57, 37, 2], and architectures [1, 45, 40].

However, despite such rapid and widespread adaptation, evaluation of Mamba's ability to perform standard downstream learning abilities exhibited by Transformer-based LLMs have either not been extensively conducted on *standard natural benchmarks* or are completely lacking. For instance, while recent works [45, 19, 30] have evaluated Mamba's ability to perform in-context learning (ICL), such studies focused extensively on either non-natural tasks [30, 17] or non-standard benchmarks [25].

---

[*]Equal Contribution

Furthermore, evaluation of Mamba's mixed-precision fine-tuning (MPFT) and performance efficient fine-tuning (PEFT) capabilities are currently lacking. For the former, MPFT (and, by extension, mixed-precision inference) are made difficult due to potential sensitivities of Mamba's recurrent dynamics, where [21, 29] suggest full precision (FP32) is required to perform stable training. For the latter, PEFT via standard low-rank adaptation (LoRA) [28] is made difficult within Mamba's SSM layer (referred to herein as the `MambaBlock`) due highly customized SSM CUDA kernels which provide competitive performance to attention-based speedups [10] at the cost of standard adapter support. However, PEFT and MPFT are arguably two of the most widely utilized techniques for LLM alignment [53] and customization [55], and are typically combined to drastically decrease hardware demands needed to fine-tune modern LLMs [12].

Herein, we extensively explore Mamba's downstream learning capabilities across standard natural benchmarks. For ICL, we show that, in contrast to recent non-standard studies showing Mamba models rival state-of-the-art (SOTA) LLMs of similar parameter counts, **the pretrained benefits of Mamba few-shot learning are significantly less than comparable Transformer LLMs across standard natural benchmarks**; averaged across the benchmarks and parameter counts in Table 1, **Mamba models only achieve 38% of the performance improvements (relative to zero-shot) of comparable Transformer models** from the Pythia suite [5]. However, we show in the sequel that **Mamba models can more than halve this gap through efficient fine-tuning**, achieving as much as 81.5% of the average few-shot learning improvement (relative to zero-shot) of comparable Transformers.

For MPFT, we leverage theory from dynamical systems to show that small input changes in a `MambaBlock` do not lead to exponentially deviating outputs. Empirically, we validate this theoretical result; compared to full-precision, deviations due to mixed-precision for Mamba inference and fine-tuning are on par with those demonstrated by Transformer LLMs (Section 6). For PEFT, we show that by targeting the largest memory buffer exploited by Mamba's highly customized CUDA kernels, LoRA may be used for extremely efficient fine-tuning, while simultaneously regularizing the majority of Mamba's SSM parameters via weight tying. We show that this leads to extremely efficient PEFT, resulting in up to 2.15 times faster training and 65.5% reduced memory compared to the largest evaluated Mamba model without MPFT or PEFT.

## 2 Background

**Downstream learning for LLMs**. Since the release of the Transformer architecture [54], attention-based LLMs have exhibited several downstream learning abilities–in particular, PEFT, MPFT, and ICL–which allow the rapid adaptation of foundation models towards specific applications. PEFT using adapters [24] allows a large pretrained model to be efficiently adapted for a particular downstream task by freezing the full model and training only a small number of extra parameters. Arguably the most widely used such PEFT method is LoRA [28], which injects trainable low-rank matrices into Transformer layers to approximate weight updates.

To further decrease the computational demands necessary for LLM fine-tuning and inference, MPFT via mixed-precision (i.e., FP16 or BF16) [31, 43] and quantized low-precision [12] have proven effective strategies to reduce GPU memory and runtime requirements without deleterious effects on downstream performance [12, 59]. Additionally, mixed-precision approaches have paved the way for hardware-aware optimizations within the self-attention module [10], greatly mitigating the quadratic complexity of Transformer LLMs. Together, PEFT and MPFT have created a rich ecosystem with which varying combinations of these approaches may be used to meet the computational constraints of a given training system. We note that post-fine-tuning quantization approaches [13] may be further used to decrease Transformer LLM computational demands, but such approaches are not considered in this work.

ICL provides an adaptable alternative to fine-tuning. Rather than fine-tune the LLM directly, ICL augments a prompt with $n$ relevant examples (called *shots*) preceding the query of interest. Given sufficiently large models and pretraining data [8, 58], Transformer LLMs have proven adept at learning new concepts on the fly provided such few-shot prompting. However, it is worth noting that ICL inference time increases dramatically as the number of shots grows (due to self-attention's quadratic complexity) and PEFT (when possible) is known to produce more accurate downstream learning results [8, 41].

Table 1: In-context learning performance for pretrained Mamba and Pythia models. Models are collected into parameter classes for head-to-head comparison using the groupings in [22]. Model checkpoints were evaluated on all benchmarks and few-shot settings using the LM evaluation harness from Eleuther AI [16]. LAMBADA zero-shot is more effective for the model sizes considered (further discussed in [61, 8]) and thus excluded from few-shot performance averages. Highlighted in bold is the top-performing few-shot learner per benchmark and model grouping.

| Model | $N$-shot | LAMBADA ppl ↓ | LAMBADA acc ↑ | HellaSwag acc ↑ | PIQA acc ↑ | Arc-E acc ↑ | Arc-C acc ↑ | WinoGrande acc ↑ | 0-shot incr. Mean % ↑ |
|---|---|---|---|---|---|---|---|---|---|
| Mamba 130M | 0 | **16.07** | **44.3** | **35.3** | 64.5 | 48.0 | **24.2** | **44.8** | – |
| | 1 | 19.34 | 38.3 | 35.2 | 64.3 | 47.1 | 23.5 | 51.3 | -1.4 |
| | 3 | 23.13 | 35.4 | 35.1 | **65.1** | **49.0** | 24.0 | 50.7 | -0.2 |
| | 5 | 24.38 | 36.2 | 34.8 | 64.9 | 49.2 | 23.9 | 50.5 | -0.5 |
| Pythia 160M | 0 | 38.20 | 32.7 | 30.2 | 61.8 | 43.4 | 23.8 | 51.0 | – |
| | 1 | 47.21 | 28.2 | 30.6 | 62.2 | 43.4 | 23.7 | 49.3 | -0.4 |
| | 3 | 63.70 | 24.7 | 30.5 | 61.9 | 44.8 | 22.9 | 51.3 | **0.1** |
| | 5 | 66.30 | 25.3 | 30.4 | 62.6 | 43.4 | 23.1 | 50.8 | -0.2 |
| Mamba 370M | 0 | **8.14** | **55.6** | **46.5** | 69.5 | 55.0 | 27.9 | 55.5 | – |
| | 1 | 9.74 | 49.8 | 45.9 | 69.3 | 57.4 | 26.5 | 54.6 | -0.8 |
| | 3 | 10.89 | 48.5 | 46.2 | **69.6** | **58.7** | **28.5** | 53.6 | 1.0 |
| | 5 | 11.36 | 48.5 | 46.2 | 69.4 | 58.3 | 28.0 | **56.0** | 1.3 |
| Pythia 410M | 0 | 10.83 | 51.5 | 40.6 | 66.9 | 52.0 | 24.1 | 53.4 | – |
| | 1 | 12.26 | 47.1 | 40.5 | 68.0 | 53.8 | 25.6 | 52.4 | 1.8 |
| | 3 | 14.39 | 43.2 | 40.9 | 67.9 | 55.1 | 26.9 | 54.0 | **4.2** |
| | 5 | 14.62 | 44.1 | 40.8 | 68.1 | 54.6 | 26.6 | 53.4 | 3.5 |
| Mamba 790M | 0 | **6.01** | **61.7** | **55.1** | 72.1 | 61.2 | 29.6 | 56.0 | – |
| | 1 | 7.06 | 56.2 | 54.5 | **72.5** | 63.3 | 30.1 | 56.9 | 1.4 |
| | 3 | 8.05 | 54.8 | 54.2 | 72.2 | 63.4 | 31.6 | 57.2 | 2.4 |
| | 5 | 8.83 | 53.4 | 54.6 | **72.5** | **64.6** | **32.1** | **57.5** | **3.4** |
| Pythia 1B | 0 | 7.92 | 56.3 | 47.2 | 70.7 | 57.0 | 27.0 | 53.4 | – |
| | 1 | 8.99 | 51.8 | 47.3 | 70.7 | 57.1 | 28.2 | 53.4 | 1.0 |
| | 3 | 10.48 | 48.2 | 47.5 | 71.2 | 59.2 | 28.0 | 54.3 | 2.2 |
| | 5 | 10.86 | 48.4 | 47.3 | 71.4 | 58.7 | 28.4 | 53.1 | 1.9 |
| Mamba 1.4B | 0 | **5.04** | **65.0** | **59.1** | 74.2 | 65.5 | 32.9 | 58.6 | – |
| | 1 | 5.83 | 60.6 | 58.20 | **74.7** | 64.5 | 33.0 | 61.2 | -0.5 |
| | 3 | 6.62 | 58.9 | 58.8 | 73.7 | 66.1 | 34.4 | 60.9 | 0.6 |
| | 5 | 6.98 | 58.4 | 59.0 | 74.0 | **66.4** | **35.5** | 60.5 | 1.4 |
| Pythia 1.4B | 0 | 6.09 | 61.7 | 52.1 | 70.9 | 60.5 | 28.5 | 57.4 | – |
| | 1 | 6.96 | 56.3 | 52.1 | 71.4 | 62.0 | 29.5 | 57.5 | 1.4 |
| | 3 | 7.89 | 54.4 | 52.6 | 70.9 | 63.9 | 31.1 | 56.8 | 2.9 |
| | 5 | 8.02 | 54.4 | 52.8 | 71.0 | 63.2 | 31.3 | 57.8 | **3.3** |
| Mamba 2.8B | 0 | **4.23** | **69.2** | **66.2** | 75.2 | 69.7 | 36.3 | 63.4 | – |
| | 1 | 5.01 | 63.9 | 65.7 | 75.5 | 69.8 | 37.2 | 63.7 | 0.6 |
| | 3 | 5.53 | 63.0 | 65.5 | 75.2 | 70.8 | 38.1 | **64.8** | 1.6 |
| | 5 | 5.70 | 62.7 | **66.2** | **76.2** | **70.9** | **38.3** | 64.6 | 2.1 |
| Pythia 2.8B | 0 | 5.04 | 64.7 | 59.3 | 73.9 | 64.2 | 32.9 | 59.8 | – |
| | 1 | 5.66 | 60.9 | 59.4 | 73.8 | 66.8 | 34.8 | 59.0 | 1.7 |
| | 3 | 6.20 | 59.1 | 59.9 | 74.7 | 67.4 | 34.9 | 60.8 | 2.9 |
| | 5 | 6.52 | 59.1 | 60.2 | 74.5 | 67.1 | 35.0 | 61.3 | **3.1** |

State-space Models. Structured state-space sequence (S4) models [23, 14] are SSMs which leverage linear time-invariant (LTI) systems to combine the computational advantages of Transformers–i.e., highly parallelizable training–and recurrent neural networks (RNNs)–i.e., subquadratic autoregressive inference using recurrency. Within the S4 layer, an input signal is discretized and LTI parameters representing the input's latent dynamics are learned. Owing to the S4 block's latent dynamics being LTI, the S4 block's output may be thus compactly represented as a single convolution between the input and an *SSM convolution kernel* (a matrix whose entries are products of LTI learnable parameters resulting from unrolling the state-space equations). However, despite hardware efficiency and long-dependency-modeling improvements, LTI-based S4 models remained inferior to Transformers of comparable parameter-sizes for natural language tasks, even when augmenting S4 layers with attention-layers for hybrid architectures [22].

Innovating on these previous S4 approaches, Mamba utilizes time-*varying* parameters to model latent dynamics, thus broadening the ability to capture nuanced changes evolving in discrete-time. Without LTI dynamics, however, the input-output representation via the SSM convolution kernel is no longer applicable, thus voiding previous hardware-aware S4 optimizations [14]. To enable hardware efficiency with time-varying SSM parameters, [22] thus introduced extensively customized CUDA kernels which implement highly parallelized prefix sums to compute recurrent states.

# 3   Mamba state-space models

For model dimension $d$ and maximum input sequence length $T$, the `MambaBlock` defines state-space parameters $\mathbf{A}, \mathbf{B}_t, \mathbf{C}_t, \boldsymbol{\Delta}_t \in \mathbb{R}^{d \times d}$ for $t \in \{1, \dots, T\}$. The matrix $\boldsymbol{\Delta}_t$ controls the discrete step-size. Given an input sequence $\mathbf{u}_1, \dots, \mathbf{u}_T \in \mathbb{R}^d$, the following linear mapping through latent states $\boldsymbol{x}_1, \dots, \boldsymbol{x}_T \in \mathbb{R}^d$ is used to produce the output $\mathbf{y}_1, \dots, \mathbf{y}_T \in \mathbb{R}^d$:

$$\boldsymbol{x}_t = \bar{\mathbf{A}}_t \boldsymbol{x}_{t-1} + \bar{\mathbf{B}}_t \mathbf{u}_t \tag{1}$$

$$\mathbf{y}_t = \bar{\mathbf{C}}_t \boldsymbol{x}_t, \tag{2}$$

where $\bar{\boldsymbol{\Delta}}_t = \mathtt{softplus}(\mathtt{Linear}(\boldsymbol{\Delta}_t)) \in \mathbb{R}^{d \times d}$, $\bar{\mathbf{A}}_t = \exp\left(\bar{\boldsymbol{\Delta}}_t \mathbf{A}\right)$ and $\bar{\mathbf{B}}_t = \mathbf{A}^{-1}(\bar{\mathbf{A}} - \mathbf{I})\mathbf{B}_t$. In practice, $\mathbf{A}, \mathbf{B}_t, \mathbf{C}_t$ and $\boldsymbol{\Delta}_t$ are diagonal matrices.

**Hardware-aware optimizations**. As matrices $\mathbf{B}_t, \mathbf{C}_t$ and $\boldsymbol{\Delta}_t$ are time-varying, S4 optimizations via the SSM convolution kernel [11] are no longer applicable. However, by diagonality, each dimension may be computed in parallel. Furthermore, the recurrence along every dimension is a prefix sum (also called a *scan*), which is highly parallelizable [7]. [15] thus capitalizes on this through extensively customized CUDA kernels wherein the majority of temporal variables are carefully laid out in a large buffer of GPU memory and manipulated. Instantiated as a `PyTorch` linear layer's weight matrix, this memory buffer $\mathbf{W} \in \mathbb{R}^{n \times 3d}$ is used to store and access the diagonal elements of $\mathbf{B}_t, \mathbf{C}_t$ and $\boldsymbol{\Delta}_t$ for all $t \in \{1, \dots, T\}$, such that

$$\mathbf{W}[t-1, :d] = \mathtt{diag}(\boldsymbol{\Delta}_t), \mathbf{W}[t-1, d:2d] = \mathtt{diag}(\mathbf{B}_t), \mathbf{W}[t-1, 2d:3d] = \mathtt{diag}(\mathbf{C}_t), \tag{3}$$

where $\mathbf{W}[0, :d] = \mathtt{diag}(\boldsymbol{\Delta}_1), \mathbf{W}[n-1, d:2d] = \mathtt{diag}(\mathbf{B}_T)$, and so on.

The customized Mamba prefix scan kernel heavily relies on this memory layout to optimize the access pattern of $\mathbf{W}$ in Equations 5 and 6. We note that, rather than adjusting Mamba's low-level CUDA kernels themselves to integrate LoRA within the highly optimized prefix scan, we can instead directly target $\mathbf{W}$. Doing so, we have the following, where the proof is available in Appendix A.

**Theorem 1.** *Consider the weight matrix $\mathbf{W}$ of a `MambaBlock` from Equation 3. Targeting $\mathbf{W}$ for LoRA during fine-tuning ties adaptation weights across $\mathbf{B}_t, \mathbf{C}_t$ and $\boldsymbol{\Delta}_t$.*

# 4   Stable dynamics in the `MambaBlock`

The Mamba foundation models were pretrained in full `FP32` precision. Consequently, official Mamba implementations have cautioned against fine-tuning or training in reduced precision [21, 29], with potential sensitivities of `MambaBlock` recurrent dynamics remaining an open question. We answer the latter using theory from dynamical systems. For Mamba's discrete dynamic system in Equations 5 and 6, define

$$\boldsymbol{x}_t = F_\theta(\boldsymbol{x}_{t-1}, \mathbf{u}_t), \tag{4}$$

where $\theta$ denotes the time-varying parameters described in Section 3. For input sequence $\mathbf{u}_1, \ldots, \mathbf{u}_T$ and initial latent state vector $\boldsymbol{x}_0$, we thus write

$$\boldsymbol{x}_T = F_\theta(F_\theta(\ldots F_\theta(\boldsymbol{x}_0, \mathbf{u}_1))) \coloneqq F_\theta^{T-1}(\boldsymbol{x}_0, \mathbf{u}_1).$$

The rate of divergence between two scalar $\varepsilon$-close inputs to a discrete dynamical system is bounded by the system's maximal Lyapunov exponent $\lambda_{\mathtt{max}}$ [44]. Given $\lambda_{\mathtt{max}}$ and two initial values $(\boldsymbol{x}_0, \mathbf{u}_1)$ and $(\boldsymbol{x}_0 + \varepsilon, \mathbf{u}_1 + \varepsilon)$, the maximum deviation between these points grows as [33, 50]:

$$\max |F_\theta^N(\boldsymbol{x}_0, \mathbf{u}_1) - F_\theta^N(\boldsymbol{x}_0 + \varepsilon, \mathbf{u}_1 + \varepsilon)| \in \mathcal{O}(\varepsilon \exp{(N\lambda_{\mathtt{max}})}).$$

Thus, when $\lambda_{\mathtt{max}} > 0$, nearby trajectories exponentially separate and, when $\lambda_{\mathtt{max}} \leqslant 0$, nearby trajectories ultimately converge to the same fixed point or periodic cycles.

The maximal Lyapunov exponent is defined as

$$\lambda_{\mathtt{max}} \coloneqq \lim_{T \to \infty} \frac{1}{T} \log \left\| \prod_{t=0}^{T} \frac{\partial \boldsymbol{x}_t}{\partial \boldsymbol{x}_{t-1}} \right\|_2,$$

where $\|\|_2$ denotes the spectral norm for matrices. For an arbitrary `MambaBlock`, we prove the following:

**Theorem 2.** *Let $(\boldsymbol{x}_{t-1}, \mathbf{u}_t)$ be the latent state and input at an arbitrary time $t \in \{1, \ldots, T\}$ within a* `MambaBlock`. *Then small changes $(\boldsymbol{x}_{t-1} + \varepsilon, \mathbf{u}_t + \varepsilon)$ produce deviations which are exponentially non-increasing over discrete-time. That is,* $\max |F_\theta^N(\boldsymbol{x}_{t-1}, \mathbf{u}_t) - F_\theta^N(\boldsymbol{x}_{t-1} + \varepsilon, \mathbf{u}_t + \varepsilon)| \in \mathcal{O}(\varepsilon \exp{(N\zeta)})$, *for some scalar $\zeta \leqslant 0$.*

The proof of Theorem 2 is available in Appendix B, where the maximal Lyapunov exponent for an arbitrary `MambaBlock` is first proven to be non-positive. The main result subsequently follows.

**Consequences for automatic mixed-precision**. During a forward pass, automatic mixed-precision (AMP) saves time and memory by computing forward activations in half-precision (`FP16` or `BF16`). During a backward pass, AMP computes gradients in half-precision and up-casts to full-precision prior to updating. In contrast to full-precision fine-tuning, MPFT within the `MambaBlock` thus results in small differences to the inputs $\mathbf{u}_1, \ldots, \mathbf{u}_T$ fed into the SSM scan (which are passed through a `SwiGLU`), $\bar{\boldsymbol{\Delta}}_t$ (which is passed through a `softplus`), and the gradients calculated during training.

For a discrete dynamical system with $\lambda_{\mathtt{max}} > 0$, changes due to AMP compound after repeated expansion of the recurrent state, thus leading to exponential deviations between quantities calculated using mixed- versus full-precision. We note that Transformers are not recurrent, and thus not susceptible to such issues. Yet, just as differences introduced by quantization/mixed-precision produce output differences in Transformer results, differences are expected in Mamba results using different precision strategies. However, by Theorem 2, such differences do not exponentially compound over discrete-time within the `MambaBlock`.

# 5  Related Work

Several recent works [45, 19, 30, 40] have studied Mamba's ability to perform ICL. However, none of these have extensively studied Mamba's ICL capabilities either on standard NLP benchmarks or on pure `MambaBlock` foundation models. In particular, foundational Mamba models' ICL abilities were tested in [45] to learn simple function classes (e.g., logistic regression and decision trees [17]) and in [19] to learn non-standard NLP benchmarks (i.e., task vectors [25]). While [45, 19] report Mamba's ICL abilities rival SOTA Transformers, their utilized benchmarks were proposed as supplemental ICL studies after Transformer LLMs' success on standard NLP benchmarks [8]. Indeed, direct evaluation of Mamba foundation models on standard NLP benchmarks does not lead to higher gains over zero-shot performance relative to comparable Transformer LLMs (demonstrated in Table 1).

Lyapunov exponents have previously been considered for classic RNN structures (e.g., vanilla RNNs, LSTMs, GRUs, PLRNNs, etc.) [44, 56], to determine when such models exhibit chaotic dynamics and the impact on the exploding/vanishing gradient phenomena[*]. For more recent S4 neural

---

[*]We note that this continues a long line of research exploring RNNs sensitivity to initial conditions and their subsequent ability to produce chaotic output [47, 34, 3, 4], although previous work did not leverage Lyapunov exponents.

175 models, [18] used Hurwitz matrices to characterize the numerical stability of linear time-invariant
176 (LTI) S4 models. However, such analysis is not applicable to time-varying models, such as Mamba,
177 nor does it characterize the effects of sensitive dependence on initial conditions (e.g., divergence of
178 two $\varepsilon$ close inputs). To the best of our knowledge, no previous works have used Lyapunov exponents
179 to explore the effects of mixed-precision on recurrent neural models or Mamba architectures.

180 As in [22], the majority of subsequent Mamba works have focused on pretraining `MambaBlocks` using
181 full precision [65, 62, 1, 40]. Notably, the official implementation of Jamba [40], the Transformer-
182 Mamba hybrid, supports mixed- and 8-bit precision, but avoids `MambaBlocks` when applying such
183 quantization [32]. Similarly, the official Mamba sources advise using full precision within the
184 `MambaBlock` [29, 21], cautioning against using mixed-precision due to potential recurrent sensitivities.
185 To the best of our knowledge, no existing works have either theoretically explored the effects small
186 input changes (e.g., due to mixed-precision) have on Mamba's recurrent dynamics, empirically
187 explored such effects downstream impact on fine-tuning and inference, or explored pure Mamba
188 networks fine-tuning abilities relative to Transformer LLMs.

# 6 Experiments

190 To demonstrate the implications of Theorem 2, we explore the performance difference between
191 running inference with full-precision pretrained weights and using mixed-precision (FP16 and BF16)
192 weights. **Model performance is measured as percent accuracy** using the MMLU [26] dataset.
193 The difference in model performance is reported as the mean *divergence* (i.e., absolute difference)
194 between the original full-precision and respective mixed-precision model, averaged over {0, 1, 3,
195 5}-shot percent accuracy. Thus, **a divergence greater than one denotes an average difference**
196 **greater than one entire percentage of accuracy.**

197 Mamba pretrained checkpoints are compared to pretrained Transformer models of similar parameter
198 counts and no more than $\sim$300B total pretraining tokens (Pythia [5], OLMo [20] 336B-token
199 checkpoint, and Phi 1.5 [39]). We note that Pythia and Mamba models were both pretrained using
200 the same corpus [15], allowing the fairest comparison between SSMs and Transformers. To limit
201 extraneous numerical effects within experiments (e.g., due to parameter aggregation across multiple
202 GPUs), all models were run using a single GPU (Nvidia A10G, 24 GB total memory). All models
203 were evaluated using the LM evaluation harness from Eleuther AI [16]. Further experimental details
204 are available in Appendix C. The results are available in Table 2.

Table 2: Mean full-precision (FP32) divergence in MMLU performance for mixed-precision inference.
Divergence is averaged over {0, 1, 3, 5}-shot performance. Pretrained checkpoints are used for
Mamba (M), Pythia (P), OLMo [20], and Phi-1.5 [39] (Phi) models.

| Model | M | P | M | P | M | P | OLMo | M | P | Phi | M | P |
|---|---|---|---|---|---|---|---|---|---|---|---|---|
| Size | 130m | 160m | 370m | 410m | 790m | 1b | | 1.4b | | 1.5b | 2.8b | |
| FP16 $\mu$ | 0.03 | 0.35 | 0.05 | 0.06 | 0.21 | 0.05 | 0.04 | 0.04 | 0.07 | 0.03 | 0.15 | 0.12 |
| BF16 $\mu$ | 0.05 | 1.45 | 0.20 | 0.20 | 0.66 | 0.16 | 0.13 | 0.31 | 0.13 | 1.05 | 1.17 | 0.11 |

205 From Table 2, inferencing in Pythia using `FP16` and `BF16` result in an average 0.13 and 0.41 full-
206 precision divergence, respectively. Mamba displays similar averages in comparison: inferencing in
207 Mamba using `FP16` and `BF16` result in an average 0.10 and 0.48 divergence, respectively. Interestingly,
208 both SSM and Transformer architectures exhibit *large divergence spikes*–i.e., mean divergence greater
209 than a percentage point–when using `BF16`, which occurs once for Mamba and Phi 1.5 models and
210 twice for Pythia models. In the following, we show that such spikes may be mitigated for Mamba
211 SSMs by combining mixed-precision with parameter-efficient adapters during fine-tuning.

212 **Non-divergent Mamba fine-tuning.** We next explore the implications of Theorem 2 on fine-tuning,
213 wherein mixed-precision is especially critical; MPFT combined with PEFT adapters have been shown
214 to drastically reduce Transformer fine-tuning times [12]. We are thus interested in the divergence
215 between Mamba models fully fine-tuned (i.e., no adapters, all model weights are trained) in full-
216 precision and models fine-tuned using mixed-precision and/or PEFT adapters. We focus on utilizing
217 LoRA [28], which is arguably the most widely used PEFT framework for LLMs.

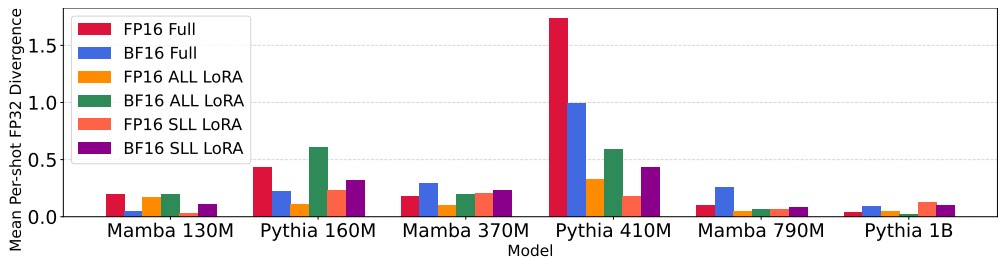

Figure 1: Mean full-precision (FP32) divergence in MMLU performance for Mamba and Pythia models. Models are fine-tuned over the `Alpaca` dataset [51] using different combinations of MPFT and PEFT. Full fine-tuning (i.e., no PEFT adapters) is denoted as `Full`.

Using the `Alpaca` dataset [51], `Mamba` 160M, 410M, and 790M models are fine-tuned for three epochs with a maximum sequence length of 512. We denote the targeting of all linear layers (ALL) for LoRA as *ALL LoRA*, the targeting of a subset of linear layers (SLL) for LoRA as *SLL LoRA*, and no adapters as *Full* (i.e., full fine-tuning). Both `ALL` and `SLL LoRA` adapt the large memory buffer described in Theorem 1.

Each fine-tuning run occurred on a single A10G GPU. To further limit extraneous numerical effects, the same batch size is used for all FP32, FP16, and BF16 experiments for a given model size. While this leads to hardware underutilization (i.e., non-saturated GPU memory for mixed-precision and LoRA experiments), this is necessary to guarantee no divergence is due to differences in parameter update schedules. For comparison, `Pythia` 160M, 410M, and 1B models are fine-tuned using the same experimental setup. The training recipe for all models was adapted from [53], with the `AdamW_torch` optimizer and a `cosine annealing` schedule. Further experimental details are available in Appendix C.

For each Mamba and Pythia model, Figure 1 shows the mean divergence calculated between the respective FP32 `Full` and mixed-precision ALL/SLL LoRA fine-tuned models, averaged over {0, 1, 3, 5}-shot MMLU accuracy. Across mixed-precisions and adapter settings, Mamba displays comparable divergences to Pythia models. E.g., **for** `FP16`**, Mamba demonstrates an average divergence of 0.1, compared to 0.14 for Pythia**. Similarly, for `BF16`**, Mamba demonstrates an average divergence of 0.18, compared to 0.28 for Pythia**. Importantly, Mamba models do not exhibit large deviation spikes after fine-tuning (in contrast to Pythia models).

**Hardware throughput and memory-utilization improvements**. With comparable divergences to Transformers and stable dynamics, we show that MPFT and PEFT may be used to significantly increase GPU-training throughput for Mamba SSMs. To demonstrate such improvements, we utilize the previous fine-tuning settings for the Alpaca dataset. However, we now adjust the batch size to maximize throughput per MPFT and PEFT configuration.

For each MPFT and PEFT configuration, the *average tokens-per-second* (ATPS) is calculated as the total tokens used for fine-tuning divided by total training time, and the *maximum memory-per-token* (MMPT) is calculated as the maximum GPU memory utilization incurred (over the entire fine-tuning run) divided by the total number of tokens in each mini-batch. Results are plotted in Figure 6.

Both throughput and memory utilization improve as the number of Mamba parameters increases in Figure 6. **Compared to the full-precision full fine-tuning of** `Mamba` 790M (the largest model supported by an `A10G`'s memory capacity), evaluated **MPFT and PEFT combinations result in an average 2.15 times more training tokens-per-second while reducing per-token memory utilization by an average 62.7%**. Across all model sizes, evaluated MPFT and PEFT combinations result in an average 1.74 times more training tokens-per-second while reducing per-token memory utilization by an average 47.2% compared to respective full-precision fine-tuned runs.

### 6.1 Fine-tuning narrows the ICL gap between Mamba and Transformers

We next explore how MPFT and PEFT affect Mamba ICL performance. All Mamba pretrained models are instruction fine-tuned using `ALL LoRA` and the OpenHermes dataset [52] (which consists of 242,000 supervised samples). We use the training recipe of [53], which includes `BF16` utilization.

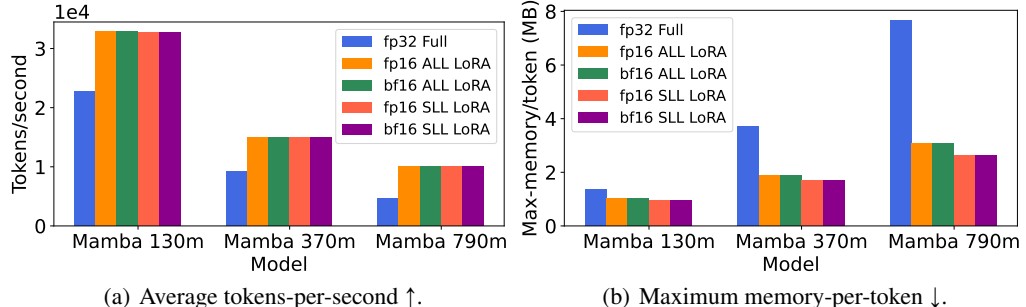

(a) Average tokens-per-second ↑.

(b) Maximum memory-per-token ↓.

Figure 2: Timing and memory usage calculated Mamba model-sizes and PEFT combinations. Each model was trained using the Alpaca dataset [51] dataset for three epochs and maximum sequence length 512. For each PEFT combination, the batch size was tuned to maximize GPU occupancy.

Performance is evaluated using the datasets from Table 1–HellaSwag [64], PIQA [6], Arc-E [9], Arc-C [9], and WinoGrande [49]–and report the *average improvement percentage* of $\{1, 3, 5\}$-*shot* versus 0-*shot* (AIPSS). For comparison, Pythia pretrained models are instruction fine-tuned using the same training recipe and `ALL LoRA` (i.e., all Pythia linear layers are adapted).

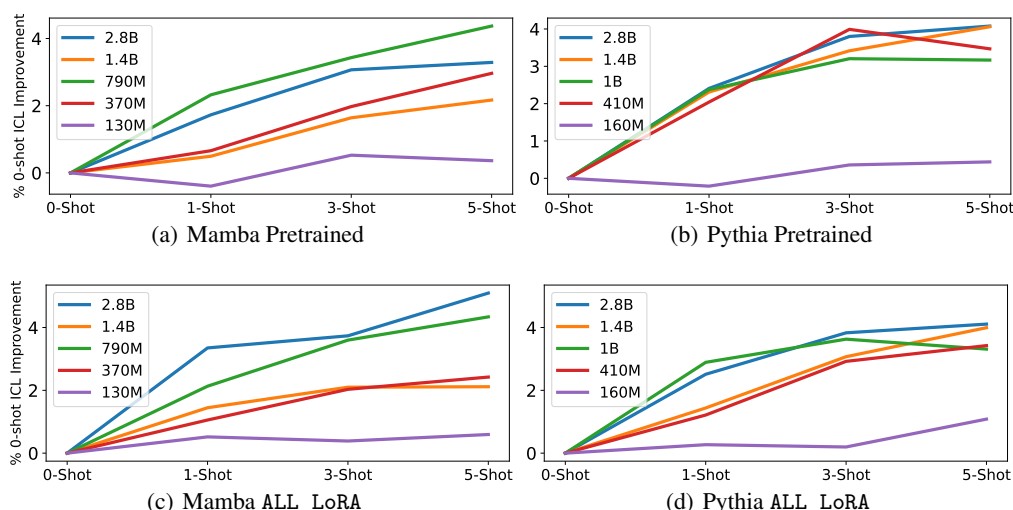

(a) Mamba Pretrained

(b) Pythia Pretrained

(c) Mamba `ALL LoRA`

(d) Pythia `ALL LoRA`

Figure 3: Fine-tuning narrows the ICL gap between Mamba and Pythia. `ALL LoRA` models were instruction fine-tuned on the OpenHermes [52] dataset for one epoch. Performance is reported as the average improvement percentage of $\{1, 3, 5\}$-shot versus 0-shot over five standard benchmarks.

Figure 3 displays AIPSS for pretrained and instruction fine-tuned Mamba and Pythia models. As previously noted, pretrained Mamba models do not display similar ICL ability as comparable Pythia models on the evaluated standard NLP benchmarks. In particular, `Mamba 2.8B`, the largest pretrained Mamba model, displays inconsistent zero-shot improvements as the number of shots increase. However, after fine-tuning, all Mamba models larger than `Mamba 130M` consistently improve in ICL performance as the number of shots increase. Compared to Mamba pretrained models, which are only capable of 38% of the AIPSS compared to similar pretrained Pythia models, fine-tuned `ALL LoRA` Mamba models are capable of 81.5% of the AIPSS compared to similarly fine-tuned Pythia models.

**Fine-tuning robustness**. We show that Mamba is robust to the choice of PEFT hyperparemters. We conduct an extensive hyperparameter search across the learning rate, LoRA dimension, and number of warmup steps. From the Cartesian-product of these three parameters, 150 hyperparameter configurations were sampled and used to fine-tune `Mamba 370M` over the Openhermes dataset. For comparison, `Pythia 410M` is similarly fine-tuned using the same set of 150 hyperparameter configurations.

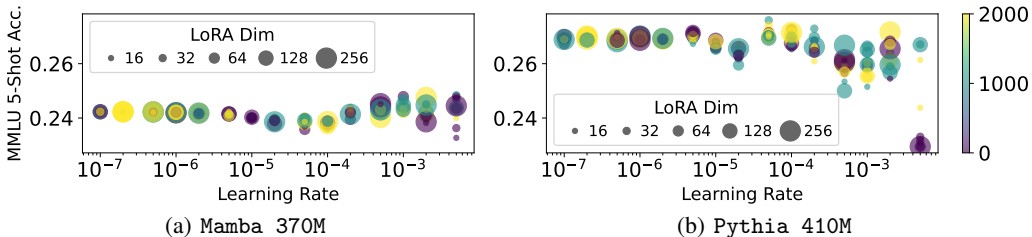

(a) `Mamba 370M`    (b) `Pythia 410M`

Figure 4: Fine-tuning hyperparameter search for OpenHermes. Each point is a different hyperparameter configuration. `SLL LoRA` was used for both models. The $x$-axis is the learning rate, the $y$-axis is resulting MMLU 5-shot performance, bubble size is the LoRA dimension, and the color is the number of warmup steps $\in \{0, 1k, 2k\}$.

The MMLU 5-shot performance for each of the 150 Mamba and Pythia fine-tuned models is displayed in 6.1. `Pythia 410M` is capable of higher performance than `Mamba 370M`, where the average accuracy for the former and the latter are 26.5% and 24.8%, respectively. However, `Mamba 370M` is much more robust to the choice of hyperparameters, with a difference of 1.5% between the minimum (23.3%) and maximum (24.8%). In contrast, `Pythia 410M` fine-tuned models display a large performance difference of 4.7% between the minimum (22.9%) and maximum (27.6%).

# 7  Discussion

We've extensively explored Mamba's downstream learning capabilities. Using dynamical systems theory, we've shown that Mamba's recurrent dynamics are robust to small input perturbations (contrary to the current understanding of Mamba's recurrent sensitivities). We've extensively confirmed this result, showing that: a) Mamba inference is robust to changes due to mixed-precision, (b) Mamba inference differences due to mixed-precision align with Transformers, (c) Mamba fine-tuning is robust to changes due to mixed-precision and PEFT, and (d) differences in downstream performance for Mamba due to MPFT and PEFT can be more robust than Transformers. Using both MPFT and PEFT, we've shown that instruction fine-tuning Mamba SSMs greatly narrows the previously observed ICL gap, going from only 38% (post pretraining) up to 81.5% (post fine-tuning) of the ICL abilities of similar Transformers. Furthermore, we've shown that combining MPFT and PEFT can more than halve training time and nearly triple memory efficiency for Mamba models.

There are significant avenues for future work. In particular, adapting Mamba's CUDA kernels to support more aggressive low-precision PEFT methods [12] would further decrease the hardware needed to train Mamba models, while providing additional speedups. Furthermore, while the largest pure Mamba model contains 2.8B parameters, the training speedups and improved memory utilization described herein may be applied to more efficiently pretrain larger pure Mamba SSMs (e.g., 7B parameters and greater), where Mamba models may better manifest emergent abilities previously displayed by Transformers (or even manifest previously unobserved abilities).

**Limitations.** While we explored the use of LoRA for Mamba models, many other PEFT adapters exist [41, 38, 27, 35]. Furthermore, while mixed-precision using `FP16` and `BF16` were explored, lower-precision methods exist [12] (which may be enabled by adapting Mamba's highly customized CUDA kernels). Both are interesting directions for future work. Finally, our timing and memory usage experiments using Alpaca did not consider the largest two Mamba models (1.4B and 2.8B) due to their exceeding A10G memory capacity for `FP32` full fine-tuning.

**Broader Impact.** The Mamba models considered are all LLMs, and thus have the same potential positive and negative societal impacts as other LLMs (e.g., hallucinations). Furthermore, fine-tuning is known to possibly erode existing LLM guardrails, and thus our methods may be adapted for this fine-tuning use case (as is the case for all PEFT and MPFT methods). However, our work improves the quality of Mamba models for downstream applications, which may be adapted for all positive LLM applications in society (e.g., personal assistants, task automation, code completion, etc.). Finally, our work decreases the computational constraints required to train and inference Mamba SSMs, which has implications for green ML (e.g., decreased $CO_2$ emissions, positive climate change impact, etc.). 410 GPU days were used to produce the results for this paper.

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

# A  Proof of weight-tying using LoRA in the `MambaBlock`

Due to the low-level nature of Mamba's prefix scan optimizations (discussed in Section 3), standard use of LoRA adapters is made difficult within Mamba's SSM-layer. E.g., while $B_t, C_t$ and $\Delta_t$ are conceptually `PyTorch` linear layers, their bundling in a contiguous memory block and careful manipulation makes appending a LoRA adapter on any of these invidiual matrices non-trivial (particularly, while respecting the highly specialized layout of each LoRA adapters targeted layer). However, we note that the overall design of the `MambaBlock`'s hardware optimizations may be leveraged to both efficiently learn the parameter-space for the majority of time-varying parameters (thus achieving PEFT) and regularize parameters during training (thus improving fine-tuning generalization).

**Theorem 1.** *Consider the weight matrix* $\mathbf{W}$ *of a* `MambaBlock` *from Equation 3. Targeting* $\mathbf{W}$ *for LoRA during fine-tuning ties adaptation weights across* $\mathbf{B}_t, \mathbf{C}_t$ *and* $\mathbf{\Delta}_t$.

*Proof.* Let $r$ be the specified LoRA dimension. Targeting this matrix for LoRA results in the adapter

$$\tilde{\mathbf{W}} = \mathbf{W} + \mathbf{W}'$$
$$= \mathbf{W} + \mathbf{U}\mathbf{V},$$

where $\mathbf{U} \in \mathbb{R}^{n \times r}$, $\mathbf{V} \in \mathbb{R}^{r \times 3d}$, and $\mathbf{W}$ is frozen during fine-tuning. Thus, for index $[i, j]$,

$$\mathbf{W}'[i,j] = \sum_{k=0}^{r-1} \mathbf{U}[i,k]\mathbf{V}[k,j].$$

Recall the form of $\mathbf{W}$:

$$\mathbf{W}[t-1,:d] = \mathtt{diag}(\mathbf{\Delta}_t), \mathbf{W}[t-1,d:2d] = \mathtt{diag}(\mathbf{B}_t), \mathbf{W}[t-1,2d:3d] = \mathtt{diag}(\mathbf{C}_t),$$

where $\mathbf{W}[0,:d] = \mathtt{diag}(\mathbf{\Delta}_1), \mathbf{W}[n-1,d:2d] = \mathtt{diag}(\mathbf{B}_T)$, and so on. For index $[t-1,j]$, we thus have

$$\tilde{\mathbf{W}}[t-1,j] = \mathbf{W}[t-1,j] + \mathbf{W}'[t-1,j]$$
$$= \mathbf{W}[t-1,j] + \sum_{k=0}^{r-1} \mathbf{U}[t-1,k]\mathbf{V}[k,j].$$

Thus, the weights $\mathbf{U}[t-1,:]$ are tied for any parameter $\tilde{\mathbf{W}}[t-1,j], j \in \{1,\dots,3d\}$, which are used to adapt parameters $\mathbf{\Delta}_1, \mathbf{B}_t$, and $\mathbf{C}_t$.

$\square$

## B  Mamba stable dynamics proof

Recall the state-space parameters and equations for the `MambaBlock`; $\mathbf{A}, \mathbf{B}_t, \mathbf{C}_t, \boldsymbol{\Delta}_t \in \mathbb{R}^{d \times d}$ for $t \in \{1, \ldots, n\} = [n]$. Given an input sequence $\mathbf{u}_1, \ldots, \mathbf{u}_n \in \mathbb{R}^d$, the following linear mapping through latent states $\boldsymbol{x}_1, \ldots, \boldsymbol{x}_n \in \mathbb{R}^d$ is used to produce the output $\mathbf{y}_1, \ldots, \mathbf{y}_n \in \mathbb{R}^d$:

$$\boldsymbol{x}_t = \bar{\mathbf{A}}_t \boldsymbol{x}_{t-1} + \bar{\mathbf{B}}_t \mathbf{u}_t \tag{5}$$

$$\mathbf{y}_t = \bar{\mathbf{C}}_t \boldsymbol{x}_t, \tag{6}$$

where $\bar{\boldsymbol{\Delta}}_t = \texttt{softplus}(\texttt{Linear}(\boldsymbol{\Delta}_t)) \in \mathbb{R}_{\geq}^{d \times d}$, $\bar{\mathbf{A}}_t = \exp(\bar{\boldsymbol{\Delta}}_t \mathbf{A})$, $\bar{\mathbf{B}}_t = \mathbf{A}^{-1}(\bar{\mathbf{A}} - \mathbf{I})\mathbf{B}_t$, and $\mathbb{R}_{\geq}$ is the set of non-negative real numbers. In practice, $\mathbf{A}, \mathbf{B}_t, \mathbf{C}_t$ and $\boldsymbol{\Delta}_t$ are diagonal matrices.

Furthermore, recall the following definitions:

$$\boldsymbol{x}_t = F_\theta(\boldsymbol{x}_{t-1}, \mathbf{u}_t)$$

where $\theta$ denotes the aforementioned time-varying parameters. For input sequence $\mathbf{u}_t, \ldots, \mathbf{u}_T$ and initial latent state value $\boldsymbol{x}_0$, we thus write

$$\boldsymbol{x}_T = F_\theta(F_\theta(\ldots F_\theta(\boldsymbol{x}_0, \mathbf{u}_1))) \coloneqq F_\theta^{T-1}(\boldsymbol{x}_0, \mathbf{u}_1).$$

We first prove that, given two scalar $\varepsilon$-close inputs to a `MambaBlock`, their deviations do not grow exponentially as the number of recurrences increases (Lemma 1). The main result in the paper is subsequently proved.

**Lemma 1.** *For input $(\boldsymbol{x}_0, \mathbf{u}_1)$ to a `MambaBlock`, small changes $(\boldsymbol{x}_0 + \varepsilon, \mathbf{u}_1 + \varepsilon)$ produce deviations which are exponentially non-increasing over discrete-time. That is, $\max|F_\theta^N(\boldsymbol{x}_0, \mathbf{u}_1) - F_\theta^N(\boldsymbol{x}_0 + \varepsilon, \mathbf{u}_1 + \varepsilon)| \in \mathcal{O}(\varepsilon \exp(N\zeta))$, for some scalar $\zeta \leqslant 0$.*

*Proof.* Firstly, we note that within the `MambaBlock`, $A$ is stored in log-space followed by a negative exponentiation prior to use. Thus, $\mathbf{A} \in \mathbb{R}_{\leq}^{d \times d}$, where $\mathbb{R}_{\leq}$ is the set of non-positive real numbers.

Recall that for the maximum deviation, we have:

$$\max|F_\theta^N(\boldsymbol{x}_0, \mathbf{u}_1) - F_\theta^N(\boldsymbol{x}_0 + \varepsilon, \mathbf{u}_1 + \varepsilon)| \in \mathcal{O}(\varepsilon \exp(N\lambda_{\texttt{max}})).$$

where the maximal Lyapunov exponent $\lambda_{\texttt{max}}$ is defined as:

$$\lambda_{\texttt{max}} \coloneqq \lim_{T \to \infty} \frac{1}{T} \log \left\| \prod_{t=0}^{T} \frac{\partial \boldsymbol{x}_t}{\partial \boldsymbol{x}_{t-1}} \right\|_2,$$

and $\|\|_2$ denotes the spectral norm for matrices.

Thus, to complete the proof, it suffices to show that $\lambda_{\texttt{max}} \leqslant 0$. Recall that $\mathbf{A}$ and $\bar{\boldsymbol{\Delta}}_t$ are diagonal. From Equation 5, we thus have

$$\lambda_{\texttt{max}} = \lim_{T \to \infty} \frac{1}{T} \log \left\| \prod_{t=0}^{T} \frac{\partial \boldsymbol{x}_t}{\partial \boldsymbol{x}_{t-1}} \right\|_2$$

$$= \lim_{T \to \infty} \frac{1}{T} \log \left\| \prod_{t=0}^{T} \exp(\bar{\boldsymbol{\Delta}}_t \mathbf{A}) \right\|_2$$

$$= \lim_{T \to \infty} \frac{1}{T} \log \left\| \exp \sum_{t=0}^{T} (\bar{\boldsymbol{\Delta}}_t \mathbf{A}) \right\|_2$$

Let $i$ be the dimension which corresponds to the output of the spectral norm, i.e., $i = \text{argmax}_{j=1,\ldots,d} \{\exp \sum_{t=0}^{T} (\bar{\boldsymbol{\Delta}}_t[j,j] \mathbf{A}[j,j])\}$. We thus have

$$\lambda_{\texttt{max}} = \lim_{T \to \infty} \frac{1}{T} \log \left\| \exp \sum_{t=0}^{T} (\bar{\boldsymbol{\Delta}}_t \mathbf{A}) \right\|_2$$

$$= \lim_{T \to \infty} \frac{1}{T} \log \exp \sum_{t=0}^{T} (\bar{\boldsymbol{\Delta}}_t[i,i] \mathbf{A}[i,i])$$

$$= \mathbf{A}[i,i] \lim_{T \to \infty} \frac{1}{T} \sum_{t=0}^{T} \bar{\boldsymbol{\Delta}}_t[i,i]$$

$\mathbf{A}[i, i]$ is non-positive and $\lim_{T \to \infty} \frac{1}{T} \sum_{t=0}^{T} \bar{\mathbf{\Delta}}_t[i, i] \geqslant 0$, since $\bar{\mathbf{\Delta}}_t[i, i] \in \ \forall t$. Thus, $\lambda_{\max} \leqslant 0$. $\quad\square$

**Theorem 2.** *Let* $(\boldsymbol{x}_{t-1}, \mathbf{u}_t)$ *be the latent state and input at an arbitrary time* $t \in [1, T]$ *within a* `MambaBlock`. *Then small changes* $(\boldsymbol{x}_{t-1} + \varepsilon, \mathbf{u}_t + \varepsilon)$ *produce deviations which are exponentially decreasing over discrete-time, i.e.,* $\max |F_\theta^N(\boldsymbol{x}_0, \mathbf{u}_1) - F_\theta^N(\boldsymbol{x}_0 + \varepsilon, \mathbf{u}_1 + \varepsilon)| \in \mathcal{O}(\varepsilon \exp(N\zeta))$, *for some scalar* $\zeta \leqslant 0$.

*Proof.* Let $\tau(t)$ be a function that maps time values such that $\tau(t) \in [1, T - t]$ and $\tau(t) = 1, \tau(t + 1) = 2, \ldots, \tau(t + T) = T - t$. Then $\mathbf{B}_{\tau(t)}, \mathbf{C}_{\tau(t)}, \mathbf{\Delta}_{\tau(t)}$ define a new `MambaBlock` with inputs $\mathbf{u}_{\tau(t)}, \ldots, \mathbf{u}_{\tau(t+T)}$ and subsequent recurrent states $\boldsymbol{x}_{\tau(t)}, \ldots, \boldsymbol{x}_{\tau(t+T)}$. Applying Lemma 1 to this `MambaBlock` with $(\boldsymbol{x}_{\tau(t)-1}, \mathbf{u}_{\tau(t)})$ completes the proof. $\quad\square$

# C   Experimental Details

All model checkpoints were evaluated on all benchmarks and few-shot settings using the LM evaluation harness from Eleuther AI [16], version 0.4.2. Pythia and Mamba `Huggingface` checkpoints were used for all inference and fine-tuning experiments, e.g., `EleutherAI/pythia-160m` and `state-spaces/mamba-130m-hf` for the smallest respective models. All fine-tuning experiments were run using package versions `Transformers 4.40.0.dev0`, `Accelerate 0.28.0`, TRL `0.8.1`, PyTorch `2.2.1+cu121`, and PEFT `0.10.0`.

For MPFT, `Flash Attention 2.0` [10] via `flash_attn 2.5.7` was used for Pythia models. For FP16 and BF16 inference results, Flash Attention 2.0 was used for both Pythia and OLMo models. For OLMo results, the 336B-token checkpoint was used by specifying `revision=step80000-tokens336B`.

Outside of the OpenHermes hyperparameter search, all Alpaca and OpenHermes fine-tuning experiments used the following training recipe (adapted from [53]): `AdamW_torch` optimizer, `cosine annealing` schedule, no gradient accumulation, maximum norm of 1.0 for gradient clipping, and no warmup steps. Training epochs used for all Alpaca and OpenHermes experiments were three and one, respectively. For both Pythia and Mamba models, the learning rate and LoRA dimension $r$ were scaled to improve performance of smaller models (per-model values listed in Table 3).

For SLL LoRA, targeted Mamba layers were {x_proj, embeddings, in_proj, out_proj}; x_proj is the large `MambaBlock` memory buffer which, when targeted by LoRA, regularizes the majority of SSM parameters during fine-tuning through weight tying (Theorem 1). Pythia targeted SLL LoRA layers were {dense, embed_in, query_key_value, dense_h_to_4h, dense_4h_to_h}, chosen to balance performance across model sizes.

All experiments in Tables 1 and 2, Figures 1 and 6 were run using a signle-GPU Nvidia A10G (24 GB total memory). For Pythia and Mamba ALL LoRA experiments in Figure 3, all experiments were run on an A10G, except for `Mamba 2.8B`, which exceeded A10G memory capacity and was run on an Nvidia H100 (80 GB total memory).

Table 3: Learning rate and LoRA dimension $r$ values

| Mamba size | Pythia size | learning rate | LoRA $r$ |
| --- | --- | --- | --- |
| 130M | 160M | 1.0e-5 | 8 |
| 370M | 410M | 5.0e-5 | 16 |
| 790M | 1B | 1.0e-6 | 32 |
| 1.4B | 1.4B | 5.0e-6 | 64 |
| 2.8B | 2.8B | 5.0e-7 | 128 |

For the hyperparameter search results in Figure 6.1, all experiments were run using 8 H100 GPUs. SLL LoRA was used for Mamba and Pythia models. The range of hyperparameter values was as follows:

- learning rate $\in \{1e-7, 2e-7, 5e-7, 1e-6, 2e-6, 5e-6, 1e-5, 2e-5, 5e-5, 1e-4, 2e-4, 5e-4, 1e-3, 2e-3, 5e-3\}$

- LoRA dimension $r \in \{16, 32, 64, 128, 256\}$
- warmup steps $\in \{0, 1000, 2000\}$

All other hyperparameters followed previous experiments.

The Alpaca dataset is freely available for download at `ttps://huggingface.co/datasets/tatsu-lab/alpaca` under open-source license `CC-by-NC 4.0`. The OpenHermes dataset is freely available for download at `https://huggingface.co/datasets/teknium/OpenHermes-2.5` under open-source license `MIT, Apache 2.0, CC`.

