# OpenReview forum: "Mamba State-Space Models Can Be Strong Downstream Learners"
_NeurIPS.cc/2024/Conference — Submitted to NeurIPS 2024_

### Official Review · Reviewer_Uruo · 2024-07-09

**Soundness:** 2
**Presentation:** 3
**Contribution:** 3
**Rating:** 7
**Confidence:** 4

**Summary:**

This paper explores the capabilities of Mamba state-space models (SSMs) in comparison to Transformer large language models (LLMs) in various downstream learning tasks. Despite Mamba's success in some areas, the paper identifies challenges and limitations in achieving performance parity with Transformers on standard benchmarks, particularly in in-context learning (ICL), mixed-precision fine-tuning (MPFT), and parameter-efficient fine-tuning (PEFT). The study demonstrates that while Mamba models have robust recurrent dynamics and can achieve significant speed and memory efficiency gains through fine-tuning techniques, their downstream learning improvements still lag behind those of Transformers.

**Strengths:**

1. This paper is well-written and easy to read.

2. The study shows that Mamba’s recurrent dynamics are robust to small input changes, which is validated both theoretically and empirically. This robustness ensures stability in training and fine-tuning processes.

3. Despite initial shortcomings in ICL performance, extensive experiments demonstrate that mamba models exhibit strong potential for improvement through efficient fine-tuning. The models can achieve up to 81.5% of the ICL performance improvement, highlighting their adaptability with appropriate tuning methods.

**Weaknesses:**

I appreciate the authors for providing a theoretical analysis to demonstrate the controllability of implementing AMP on Mamba blocks, and the experiments indicate that PEFT is also suitable for Mamba. However, I have several concerns:

1. The authors define the Mamba process as a generalized operation: $x_t=F_{\theta}(x_{t-1},u_t)$, but the actual output of Mamba is $y_t = \bar{C_t}x_t$.Therefore, the theoretical analysis provided in the paper pertains to the stability of the hidden state under small perturbations. Is it possible to extend this analysis directly to the output $y_t$? Since the stability of the hidden state does not necessarily imply the stability of the output.
2. Theorem 1 ensures the feasibility of implementing LoRA on Mamba blocks but focuses on the $W$ matrix, neglecting the consideration of the most crucial transition matrix $\bar{A_t}$ in Mamba. Does this mean that the $\bar{A_t}$ matrix was not subjected to LoRA during fine-tuning? If so, is it possible to consider applying PEFT to the $\bar{A_t}$ matrix as well?
3. As an empirically-driven paper, would it be possible to include more backbones for comparison in future versions? Currently, the only baseline for comparison is Pythia.

**Questions:**

Please see weakness.

**Limitations:**

The authors raise some limitaions, for example, the lower-precision method is not explored in this paper, but the authors claime to solve them in the future.

---

> ### Author Rebuttal · Authors · 2024-08-06
>
> We thank the reviewer for their detailed feedback.  Please find responses to your primary concerns below.
>
> ### ***Is it possible to extend this analysis directly to the output? Since the stability of the hidden state does not necessarily imply the stability of the output.***
> We thank the reviewer for raising this interesting point.  Within the Mambablock, stability of the hidden states indeed directly implies the stability of the output states.  To see this, recall that in the Mamba state-space equations, the input at time $t$ depends on the current input and the previous hidden state.  Thus, the recurrent dynamics are uniquely determined by the hidden states and, hence, **the stability of the hidden states imply the stability of the entire system (including the outputs)**.  I.e., while $y_t$ is output, the actual state fed into the system for the next iteration is $x_t$.
>
> ### ***Is*** $\bar{\mathbf{A}}_t$ ***learned during LoRA fine-tuning?***
> We thank the reviewer for raising this question.  Indeed, the matrix $\bar{\mathbf{A}}_t$ is learned during LoRA fine-tuning.  From Equation 3 of the submission, the targeted linear layer includes $\bf{\Delta}_t$.  Within the Mambablock, we have $\bar{\mathbf{A}}_t = \exp{\bf{A \Delta}_t}$.  Thus, all time-varying Mamba matrices are fine-tuned by targeting the memory buffer $\bf{W}$ using LoRA.
>
> ### ***As an empirically-driven paper, would it be possible to include more backbones for comparison in future versions? Currently, the only baseline for comparison is Pythia.***
> We thank the reviewer for raising this point and agree, more baseline model comparisons will strengthen the paper.  Toward this end, we have added additional experiments with Apple's OpenELM-{270M, 450M, 1.1B}, TinyLlama 1.1B, and AI2's OLMo 1.2B.  New experiments include studying the fine-tuning stability under mixed-precision for OpenELM-{270M, 450M} models (Figure 1 of the PDF attached to the global rebuttal), as well as new OpenHermes fine-tuning and ICL evaluations for OpenELM-{270M, 450M, 1.1B}, TinyLlama 1.1B, and OLMo 1.2B (Figure 2 of the PDF attached to the global rebuttal).  All together, these new experiments include 23 additional fine-tuned models and 272 total additional natural language benchmark evaluations.
>
> An extensive discussion of these results is included in the global rebuttal.  In particular, for mixed-precision stability, OpenELM displayed large deviation spikes (as previously seen with Pythia models).  Thus, models from both Transformer families considered exhibit less stable fine-tuning given mixed precision compared to Mamba LLMs.
>
> For the additional ICL results, we first note the several factors which make comparison between Mamba/Pythia and other LLMs difficult.  While Pythia and Mamba models were pretrained on the same dataset as well as the same total number of pretraining tokens, other LLMs have been pretrained on newer/more complex datasets for larger pretraining total tokens--e.g., the nearest available checkpoint to 300B for OpenELM models was 429B pretraining tokens, meaning these models were trained 43% longer than Pythia/Mamba models on cleaner and larger datasets.  As ICL is an emergent ability--and hence highly dependent on data quality and number of total pretraining token counts [1]--we thus reiterate and emphasize that Pythia and Mamba models remain the most apples-to-apples comparison, while comparison between any other model may be influenced by pretraining duration and data differences (rather than architecture differences).
>
> With that said, we can see that in Figure 3 (of the global rebuttal PDF), pretrained Mamba models perform near the bottom of all other considered models (particularly true for models > 450M parameters).  However, post fine-tuning, Mamba ICL abilities regularly improve; among models > 450M parameters, the largest Mamba model displays ICL ability towards the top, while the Mamba-1.4B model moves towards the cluster of Transformer-based LLMs.
>
> Interestingly, pretrained OpenELM models display weaker ICL abilities than Pythia models (even weaker than some smaller Mamba models, when comparing OpenELM-450M and Mamba-370M).  However, post fine-tuning, OpenELM significantly improves; this is notable as the OpenELM models' pretraining data contain Pythia's pretraining dataset in addition to subsets of OLMo's and TinyLlama's pretraining datasets.  Furthermore, TinyLlama and OpenELM models utilize the same architectures (i.e.,  rotatory positional embeddings, RMSNorm, grouped query attention, SwiGLU activations, and even use of the Llama-2 tokenizer).  Yet, while OpenELM significantly improved post fine-tuning, TinyLlama did not (achieving a slight decrease in average ICL capability of -0.0011 after fine-tuning).  Thus, a significantly larger/diverse dataset and larger pretraining token counts are potential avenues to further improve the ICL capabilities of Mamba foundation models in future work.
>
> ### **References**
> [1] Wei, Jason, et al. "Emergent Abilities of Large Language Models." TMLR 2022

---

> > ### Author Response · Authors · 2024-08-10
> > **Follow-up**
> >
> > We thank the reviewer again for their insightful questions and comments.  The resulting additional baseline model evaluations further demonstrate the original observations regarding the stability of mixed-precision fine-tuning for Transformer LLMs (i.e., OpenELM models display large deviation spikes, similar to Pythia, while Mamba models do not) and Mamba ICL improvements after fine-tuning (i.e., pretrained Mamba ICL is generally poor compared to other Transformer LLMs, but improves towards the general performance of the other evaluated LLMs after fine-tuning).
> >
> > We will add these new experiments to the paper, in addition to a discussion of the stability of the Mambablock's outputs (given the stability of the hidden states) and the learning of all time-varying matrices given the targeting of the $\mathbf{W}$ matrix using LoRA.  We believe these new experiments and discussion are valuable additions to the paper.
> > Please let us know if we have not fully addressed your concerns.

---

> > > ### Comment · Reviewer_Uruo · 2024-08-12
> > > **Response to the authors**
> > >
> > > Dear authors,
> > >
> > > Thanks for your rebuttal! Now I understand that $\bar{A_t}$ will also be learned through the finetuning stage, and I appreciate that the authors supplemented extensive experiments to answer my suggestion of providing more comparisons. However, I still have some reservations about my first question. I agree that the state $h_t$ plays a key role in transmitting information in Mamba, but we always get the $y_t$ as the output feature. Therefore, I am still confused about whether the theorem in this paper can support the stability of the Mamba output. Could the authors explain this point?
> > >
> > > Best,
> > > reviewer Uruo

---

> ### Author Response · Authors · 2024-08-12
> **Proof of the output stability given stability of the latent states**
>
> We thank the reviewer for the healthy discussion.  We agree that this result is not immediately obvious.  We've included a rigorous proof which shows that stability of the output states (in the Mambablock) follow given the stability of the latent states.  Note that the notation below differs slightly from the paper (Mathjax would not render properly with too many mathbfs and overbars).  We believe this result will make a strong addition to the supplementary, please let us know if we've clarified your question.
>
> ### **Stable Output dynamics**
> Assume small changes $(x_{t-1} + \varepsilon,u_t + \varepsilon)$ produce deviations which are exponentially non-increasing over discrete-time.  Then small changes to the output $y_t$ are also exponentially non-increasing over discrete-time.
>
> Proof:
> Recall that $x_T = F_{\theta}^{T-1}(x_{0}, u_1)$.  Furthermore, recall that from the Mamba state-space equations:
> $$ y_t = C_t x_t $$
> where $C_t$ is a diagonal matrix.
>
> Let $ y_T = G_{\theta}^{T-1}(x_{0}, u_1) =C_T x_T = C_T F_{\theta}^{T-1}(x_{0}, u_1)$.
>
> Consider $\varepsilon$-close inputs $(x_{t-1},u_t)$ and $(x_{t-1} + \varepsilon,u_t + \varepsilon)$ and their respective outputs $y_t$ and $y_t'$.  Assume small changes $(x_{t-1} + \varepsilon,u_t + \varepsilon)$ produce deviations which are exponentially non-increasing over discrete-time.  That is,  $\max | F_{\theta}^N( x_{0}, u_1) - F_{\theta}^N( x_{0} + \varepsilon, u_1+ \varepsilon)| \in O( \varepsilon \exp{(N \zeta )} )$, for some scalar $\zeta \leq 0$.
>
> We thus have
> $$\max | y_t - y_t' | = \max | G_{\theta}^N( x_{0}, u_1) - G_{\theta}^N( x_{0} + \varepsilon, u_1+ \varepsilon)| =
> \max | C_N F_{\theta}^N( x_{0}, u_1) - C_N F_{\theta}^N( x_{0} + \varepsilon, u_1+ \varepsilon)|\\\\
> \propto \max | F_{\theta}^N( x_{0}, u_1) - F_{\theta}^N( x_{0} + \varepsilon, u_1+ \varepsilon)|\
> $$
> where proportionality follows due to the diagonality of $C_N$ and the vector-absolute value.
> Thus,
> $$\max | G_{\theta}^N( x_{0}, u_1) - G_{\theta}^N( x_{0} + \varepsilon, u_1+ \varepsilon)|  \in O( \varepsilon \exp{(N \zeta )} ),$$
> for some scalar $\zeta \leq 0$.
> $\square$

---

> > ### Comment · Reviewer_Uruo · 2024-08-13
> > **Response to the authors**
> >
> > Dear authors,
> >
> > Great, this is consistent with the proof I envisioned. Please include a discussion of the positive and negative conditions of $\zeta$ in a future version, similar to the discussion of $\lambda_{max}$ in the original article.
> >
> > To sum up, I think the authors did a good job of solving all my concerns, so I improved my score. Please add necessary parts to the future version to improve the quality of the article, thanks.
> >
> > Best,
> > reviewer Uruo

---

> > > ### Author Response · Authors · 2024-08-13
> > > **Official comment by the Authors**
> > >
> > > We will add the discussion of $\zeta$ to future versions of the paper, along with the additional experiments and other noted changes.
> > >
> > > We thank the reviewer once again for their helpful feedback and professionalism, the paper has greatly benefited from the positive discussion.

---

### Official Review · Reviewer_32SR · 2024-07-11

**Soundness:** 3
**Presentation:** 3
**Contribution:** 2
**Rating:** 4
**Confidence:** 3

**Summary:**

This paper explores Mamba's downstream learning capabilities through two primary aspects: (i) fine-tuning and (ii) in-context learning. Specifically, it examines the training stability and robustness of fine-tuning when mixed precision is applied, as well as Mamba's ability to perform in-context learning. The contributions of this paper include:

- Theoretical analysis of the stable dynamics of Mamba.
- The theoretical analysis is corroborated by the experiments.
- Experimental demonstration of Mamba's limitations on real datasets in terms of ICL.
- ICL performance improvement of Mamba through fine-tuning.

**Strengths:**

- The theoretical analysis section, although I did not verify the proofs, offers valuable insights and paves the way for more complex analyses in future work.
- This paper is well-motivated.

**Weaknesses:**

The weakness is majorly from the ICL part.

In fact, the authors show that pretrained Mamba cannot learn well via ICL, but can learn well after fine-tuning. This fact indicates that this limitation does not come from Mamba architecture itself, which is also consistent with the observation of other works such as [1]. Therefore, the limitations observed in this paper is just a general limitation caused by training recipes, which is not Mamba-specific and has been studied in many works. The solution is also standard, and the improvements are also expected since once trains well, Mamba should be able to perform in-context learning as shown in [1].

Moreover, many questions are still unclear. For instance, why does Mamba suffer from such limitations?

Therefore, the study in terms of the ICL part is lack of depth, novelty, and technical contribution.

In terms of the mixed precision part, there are also many places that are unclear to me. I suggest the authors to use more space in discussing the speciality of Mamba compared to Transformers, and what structure of Mamba caused this problem. If recurrence is the main cause of the problem, having more experiments of similar models such as GLA, linear attention, etc would also help readers to understand more about the phenomenon.

---

References

[1] Park, Jongho, et al. "Can mamba learn how to learn? a comparative study on in-context learning tasks." ICML 2024.

**Questions:**

- Is mean divergence the main metric in measuring the performance difference of a model in terms of different precision?
- Where does the numbers in L267 and L269 comes from? Based on Figure 3, it looks like the percentage of improvements should be less than 5% by observing the y limit of the figures.

**Limitations:**

The work is well-motivated, but the study lacks depth.

---

> ### Author Rebuttal · Authors · 2024-08-06
>
> We thank the reviewer for their time and review.  We first address the reviewer's main claims from their review, then their additional questions.
>
> ### ***The solution is also standard, and the improvements are also expected since once trains well, Mamba should be able to perform in-context learning as shown in [1].***
> We first point out that **the study in [1] did not evaluate a single Mamba pretrained model, nor fine-tune a single Mamba model**.  As stated in [1]:
> >we demonstrate that Mamba can be trained from scratch to perform various ICL tasks
>
> Thus, **fine-tuning to improve ICL for Mamba models (as is done in our submission) has not been previously considered**, and the work in [1] does not imply this is a given solution.  Furthermore, the work in [1] does not immediately generalize to natural language tasks; as pointed out in our submission (line 31 and lines 165-167), **[1] did not consider a single natural language task**.  **Thus, any conclusions do not readily apply to natural language, as clearly stated in [1]**:
> >Future research directions include exploring (1) how the performance on our ICL suite correlates with general language modeling capabilities, such as perplexity on standard NLP benchmarks.
>
> To further clarify, [1] considers the framework in [2], where a given model is trained (from scratch) per simple function class to study non-NLP ICL.  As discussed in [3,4], we note that the majority of LLM use cases involve natural language tasks.
>
> ### ***Therefore, the limitations observed in this paper is just a general limitation caused by training recipes***
> This is inaccurate; if this were true, pretrained Pythia, from which Mamba's pretraining recipe and data were derived from, would also not exhibit ICL.  Yet, **as seen in the paper, pretrained Pythia models excel at ICL, whereas pretrained Mamba models struggle**.
>
> Furthermore, the hyperparameter search included in our submission (Figure 4, lines 270-280) shows that Mamba is at least as robust to fine-tuning hyperparameters as Pythia. This experiment covers hundreds of fine-tuning runs and demonstrates that the main ICL results are not merely the consequence of specific hyperparameters.
>
> ### ***The weakness is majorly from the ICL part.***
> As stated throughout the submission (e.g., line 10), **two novel goals of the paper are to understand the amenability of MPFT and PEFT for Mamba architectures**.  We provided theoretical results proving Mamba models are highly amenable towards MPFT and PEFT, and verified these theoretical results through extensive experiments.  Exploring ICL given these results is yet another contribution, however, we ask the reviewer to please consider all contributions as a whole when evaluating the submission.
>
> ## References
> [1] Park, Jongho, et al. "Can mamba learn how to learn? a comparative study on in-context learning tasks." ICML 2024.
>
> [2] Garg, Shivam, et al. "What can transformers learn in-context? a case study of simple function classes." Advances in Neural Information Processing Systems 35 (2022): 30583-30598.
>
> [3] Naveed, Humza, et al. "A comprehensive overview of large language models." arXiv preprint arXiv:2307.06435 (2023).
>
> [4] Chang, Yupeng, et al. "A survey on evaluation of large language models." ACM Transactions on Intelligent Systems and Technology 15.3 (2024): 1-45.
>
> # Other questions:
> ***Is mean divergence the main metric in measuring the performance difference of a model in terms of different precision?***
>
>  The reviewer is correct.  In particular, we use the mean divergence between full- and mixed-precision model performance, where the average is over {0, 1, 3, 5}-shot performance (as discussed in lines 193-196).  We point out that this means for every bar in Figure 1, this thus requires:
> - Fine-tuning a model under the given MPFT + PEFT configuration
> - Fine-tuning of the full-precision model
> - Evaluation of the fine-tuned full-precision and MPFT + PEFT models over {0, 1, 3, 5}-shots on the natural language MMLU benchmark
>
> Thus, the original plot in **Figure 1 displays the information of 54 fine-tuned models and 216 natural language task evaluations**.
>
> ***Where does the numbers in L267 and L269 comes from? Based on Figure 3, it looks like the percentage of improvements should be less than 5% by observing the y limit of the figures.***
>
> The numbers in lines 268-269 are Mamba pretrained/fine-tuned AIPSS relative to Pythia pretrained/fine-tuned AIPSS (respectively).  I.e.: sum(Mamba pretrained/finetuned AIPSS) / sum(Ptyhia pretrained/finetuned AIPSS)
>
> ***In terms of the mixed precision part, there are also many places that are unclear to me.***
>
> We refer the reviewer to lines 149-161 of the submission, which contain both a broad overview of automatic mixed-precision (AMP) as well as how AMP directly impacts a MambaBlock.

---

> > ### Author Response · Authors · 2024-08-10
> > **Reminder**
> >
> > Dear Reviewer 32SR11,
> >
> > We thank you again for your efforts reviewing our submission.
> >
> > We have addressed all your concerns and look forward to your reply.
> >
> > Best regards

---

> > > ### Author Response · Authors · 2024-08-12
> > > **Looking forward to your response**
> > >
> > > Dear Reviewer 32SR,
> > >
> > > We have addressed the concerns you raised in your initial review and have submitted a detailed rebuttal. We are writing to confirm whether you have had the opportunity to review our responses. We hope that our rebuttal has addressed your questions satisfactorily and would appreciate if you could reconsider the contributions of our manuscript along with the efforts we have made during the rebuttal process.
> > >
> > > We are eager to hear your feedback and are open to any further discussion that might help clarify any remaining issues.
> > >
> > > Looking forward to your response.
> > >
> > > Best regards,
> > > Submission 12737 Authors

---

### Official Review · Reviewer_TRbW · 2024-07-15

**Soundness:** 4
**Presentation:** 3
**Contribution:** 3
**Rating:** 6
**Confidence:** 2

**Summary:**

This paper looks at improving start space models or Mamba by enabling mixed precision handling to improve inference and fine-tuning. The results show similar performance with a significantly reduced memory requirement

**Strengths:**

There are extensive results compared to full-precision models
The authors provide a proof of the theorem to back up their claim
The change to the mamba block is clear and easy to implement by others

**Weaknesses:**

The actual change is relatively minor in quantity but does deliver the author's required memory reductions.
The works don't use the larger models available due to limitations on memory requirements still

**Questions:**

nothing further

**Limitations:**

the limitations are well discussed at the end of the paper and generally relate to LLM or transformers in general too

---

> ### Author Rebuttal · Authors · 2024-08-06
>
> We thank the reviewer for their time and feedback in reviewing our paper.  We address the reviewer's main concerns below.
>
> ### ***Despite memory savings, the largest Mamba models are not evaluated due to memory limitations***
> We note, while this is true for full fine-tuning Mamba-{1.4B, 2.8B} models using the general hardware considered (i.e., an A10G with 24 GB), the MPFT and PEFT combinations explored provide enough memory savings to fine-tune these larger Mamba models (as evidence by the results in Figure 3, where the largest Mamba models were fine-tuned on the OpenHermes dataset using bf16 + ALL LoRA).
>
> To further explore the efficiency of MPFT + PEFT for the largest Mamba models, we fine-tuned the Mamba-{1.4B, 2.8B} models under four different mixed-precision + LoRA configurations using the Alpaca setup from Figure 2 of the original submission.  The **results are presented in Figure 1 of the PDF attached to the global rebuttal**. The plot demonstrates that, while time and memory requirements drastically increase when fine-tuning the larger Mamba models, the MPFT and PEFT configurations considered allow efficient training even on GPUs with as little as 24GB onboard memory.  We will add these results to future versions of the paper.
>
> We thank the reviewer for raising this point, as these results further demonstrate the practical speed and memory savings possible for even the largest Mamba models using MPFT and PEFT.
>
> ### ***The actual change is relatively minor in quantity***
> We agree that leveraging mixed-precision for Mamba models is simple to implement given the current ecosystem of deep learning and LLM packages.  However, we note that the current understanding of Mamba SSMs promotes that they require full-precision during training to ensure stability (as stated on lines 180-184 of the submission, and stated on both the Huggingface Mamba PEFT pages [1,2] and official Mamba github [3]).  Thus, a major goal of our submission is to theoretically prove that Mamba SSMs are, in fact, robust to changes caused by automatic mixed-precision (AMP), extensively demonstrate this empirically, and shed light on the improved hardware utilization possible through AMP (as well as opening the door for consideration of lower precision schemes in future work).
>
> ### **References**
> [1] https://huggingface.co/state-spaces/mamba-2.8b-hf#peft-finetuning-example
>
> [2] https://huggingface.co/docs/transformers/en/model_doc/mamba#peft-finetuning
>
> [3] https://github.com/state-spaces/mamba?tab=readme-ov-file#precision

---

> > ### Author Response · Authors · 2024-08-10
> > **Follow-up**
> >
> > We thank the reviewer again for their review and helpful comments.  We will add the new experiments addressing these comments to the next version of the paper (described above and in the global rebuttal), and discuss how MPFT + PEFT enables accurate fine-tuning of even the largest Mamba models.  We believe these additions further strengthen the paper overall.
> >
> > Please let us know if there are any other concerns to discuss.

---

> > > ### Comment · Reviewer_TRbW · 2024-08-12
> > >
> > > thank you for your response, i'm happy that these points, will improve the paper

---

### Author Rebuttal · Authors · 2024-08-06

We thank all reviewers for their feedback and time reviewing our paper.  In what follows, we summarize and address the main reviewer concerns.

# The paper's conclusions may be directly drawn from the conclusions of [1]
Our submission's presented conclusions may not be directly drawn from the conclusions of [1]; the work in [1] does not fine-tune a single LLM, evaluate a single pretrained LLM, consider MPFT or PEFT methods, or consider a single natural language task (the latter is stated on lines 6-7, 32-33, and 163-171 of our submission).  In stark contrast, our submission explores the impact of fine-tuning using MPFT and PEFT for pretrained Mamba models on natural language tasks.  Towards this end, the paper theoretically shows Mamba is suitable for both MPFT and PEFT, empirically verifies these results, and assesses performance across standard natural language tasks, where performance is measured through ICL, i.e., one of the most widely used metrics to assess the effectiveness of instruction fine-tuning.

Furthermore, we note that [1] makes it clear that conclusions on natural language tasks should not be drawn from their work:
>Future research directions include exploring (1) how the performance on our ICL suite correlates with general language modeling capabilities, such as perplexity on standard NLP benchmarks.

# The submission does not use the larger Mamba models available due to limitations on memory requirements
We thank the reviewer for raising this point.  While this is true for full fine-tuning of Mamba-{1.4B, 2.8B} models given the GPU used to perform experiments (an A10), the MPFT + PEFT configurations explored provide enough memory savings to fine-tune these larger Mamba models.  Thus, we fine-tuned the Mamba-{1.4B, 2.8B} models under four different mixed-precision + LoRA configurations using the Alpaca setup from Figure 2 of the original submission, **presented in Figure 1 of the attached PDF**.  The plot demonstrates that, while time and memory requirements drastically increase when fine-tuning the larger Mamba models, the MPFT and PEFT configurations considered allow efficient training even on GPUs with as little as 24GB onboard memory.

# Would it be possible to include more backbones for comparison in future versions? Currently, the only baseline for comparison is Pythia.
We thank the reviewer for pointing this out and agree that including more baselines for comparison will strengthen the submission.  Towards this end, we have run additional evaluations of Apple's OpenELM-{270M, 450M, 1.1B}, TinyLlama 1.1B, and AI2's OLMo 1.2B.  In order to make evaluations as equitable as possible, we focused on selecting LLMs of comparable parameter counts (i.e., not exceeding the Pythia 2.8B total parameter count) and total pretraining token counts.  While OpenELM, TinyLlama, and OLMo were pretrained on different datasets and on trillions of pretraining tokens, they offer checkpoints for pretraining-token counts closer to that of Pythia (summarized in the table below):

|Model Family | Pretraining Tokens|
---------------- | ---------------------
|Pythia       | 300B|
|Mamba        | 300B (same dataset as Pythia)|
|OpenELM      | 429B|
|TinyLlama    | 503B|
|OLMo         | 336B|

For **fine-tuning precision results**, we fine-tuned OpenELM-{270M, 450M} on the Alpaca dataset and compared MPFT+PEFT combinations to their full-precision fine-tuning counterparts (**Figure 2 of the attached PDF**).  Following the experimental set up in Figure 1 of the original submission, we evaluate each fine-tuned model on {0, 1, 3, 5}-shot learning over the MMLU natural language dataset, and report the mean divergence between a MPFT+PEFT combination and their full-precision counterpart (per-shot difference, averaged across all shots).  Thus, 18 additional fine-tuned models and 72 additional benchmark evaluations were run for this experiment.  As previously seen for Pythia models, OpenELM displays large divergence spikes.  Thus, Mamba SSMs display greater MPFT stability compared to both considered Transformer-model families.  For this experiment, where full-precision fine-tuning is required to determine mixed-precision divergence, we note that TinyLlama, OLMo, and OpenELM model sizes over 1.1B parameters exceeded memory of the hardware considered (A10G with 24 GB memory).

For **instruction fine-tuning performance**, we fine-tuned models using the OpenHermes dataset, the recipe described in Section 6.1, and the collection of natural language datasets used in Figure 3 of the original submission.  To collect these new results, 5 additional fine-tuned models and 200 additional natural language benchmark evluations were run.  **Displayed in Figure 3 of the attached PDF**, we note several important considerations when making comparisons between models: different LLMs have been pretrained on more complex datasets for larger pretraining token totals.  E.g.:
- The OpenELM pretraining data contains Pythia's pretraining dataset in addition to subsets of OLMo's and TinyLlama's pretraining datasets
- The nearest available checkpoint to 300B for OpenELM models was 429B, leading to 43% longer training than Pythia/Mamba

As ICL is an emergent ability--and hence highly dependent on data quality and number of total pretraining token counts [2]--we thus reiterate that Pythia and Mamba models remain the most apples-to-apples comparison.  Nonetheless, we can see that **while Mamba pretraining ICL is often worse than the other considered models, fine-tuning brings it's performance among that of the transformer-based LLMs**.  Interestingly, TinyLlama does not improve post fine-tuning, while OpenELM (which uses the same architecture as TinyLlama) transitions from weak pretraining ICL to strong post fine-tuning ICL.

# References
[1] Park, Jongho, et al. "Can mamba learn how to learn? a comparative study on in-context learning tasks." ICML 2024

[2] Wei, Jason, et al. "Emergent Abilities of Large Language Models." TMLR 2022

---

### Decision · Program_Chairs · 2024-09-25

**Decision:**

Reject

**Comment:**

The paper presents an exploration of Mamba's downstream learning capabilities, focusing on fine-tuning and in-context learning. The authors provide theoretical analysis. The motivation behind the study is clear and relevant, contributing to the broader understanding of model training stability and robustness, particularly with the application of mixed precision techniques.

However, there are significant concerns that need to be addressed for this work to be suitable for publication. The similarity to prior work, specifically [1], is quite pronounced, yet the paper does not cite or discuss this connection adequately. The distinction between testing in-context learning on pretrained versus unpretrained models and the distinction between synthetic tasks vs textual tasks are also very marginal, which weaken the novelty of the study. Given the results presented in [1] and the similarity to [1], the results presented in this work are more or less expected and not surprising.

Additionally, the paper lacks a clear focus, attempting to cover too many concepts at once, which muddles the central thesis and makes the overall argument difficult to follow. A more streamlined and coherent approach is necessary to improve readability and impact. Given these issues, I recommend rejecting the paper in its current form.

[1] Park, Jongho, et al. "Can mamba learn how to learn? a comparative study on in-context learning tasks." ICML 2024